# Humidified and standard oxygen therapy in acute severe asthma in children (HUMOX): A pilot randomised controlled trial

Paul S. McNamara[1]*, Dannii Clayton[2], Caroline Burchett[3], Vanessa Compton[4], Matthew Peak[5], Janet Clark[5], Ashley P. Jones[2]

1 Department of Child Health (University of Liverpool), Institute in the Park, Alder Hey Children's NHS Foundation Trust, Liverpool, United Kingdom, 2 Liverpool Clinical Trials Centre, University of Liverpool, a member of the Liverpool Health Partners, Liverpool, United Kingdom, 3 Paediatric Department, The Longhouse, Countess of Chester Hospital NHS Foundation Trust, Chester, United Kingdom, 4 Physiotherapy Department, Alder Hey Children's NHS Foundation Trust, Liverpool, United Kingdom, 5 Clinical Research Division, Institute in the Park, Alder Hey Children's NHS Foundation Trust, Liverpool, United Kingdom

* mcnamp@liverpool.ac.uk

**Data Availability Statement:** Data contain potentially identifying information and data anonymisation was not funded as part of the trial.

## Abstract

### Background

Oxygen ($O_2$) is a mainstay of treatment in acute severe asthma but how it is administered varies widely. The objectives were to examine whether a trial comparing humidified $O_2$ to standard $O_2$ in children is feasible, and specifically to obtain data on recruitment, tolerability and outcome measure stability.

### Methods

Heated humidified, cold humidified and standard $O_2$ treatments were compared for children (2–16 years) with acute severe asthma in a multi-centre, open, parallel, pilot randomised controlled trial (RCT). Multiple outcomes were assessed.

### Results

Of 258 children screened, 66 were randomised (heated humidified $O_2$ n = 25; cold humidified $O_2$ n = 21; standard $O_2$ n = 20). Median (IQR) length of stay (hours) in hospital was 37.9 (29.1), 52 (35.4) and 49.1 (29.7) for standard, heated humidified and cold humidified respectively and time (hours) on $O_2$ was 15.9 (9.4), 13.6 (14.9) and 13.1 (14.9) for the three groups respectively. The mean (standard deviation) time (hours) taken to step down nebulised to inhaled treatment was 5.6 (14.3), 35.1 (28.2) and 32.7 (20.1). Asthma Severity Score decreased in all three groups similarly, although missing data prevented complete analysis. Humidified $O_2$ was least well tolerated with eight participants discontinuing their randomised treatment early. An important barrier to recruitment was research nurse availability.

### Conclusion

Although, the results of this pilot study should not be extrapolated beyond the study sample and inferential conclusions should not be drawn from the results, this is the first RCT to

Data will be shared upon request to the Liverpool Clinical Trials Centre (email: LCTC@liverpool.ac.uk). Requests will be checked for compatibility with participant consent and the LCTC data sharing policy will be followed. Anonymised data, a copy of the annotated case report forms and protocol will be shared.

**Funding:** The trial was funded by a grant from the National Institute for Health Research for patient benefit programme (62616194). The funders had no role in study design, data collection and analysis, decision to publish, or preparation of the manuscript.

**Competing interests:** The authors have declared that no competing interests exist.

compare humidified and standard $O_2$ therapy in acute severe asthmatics of any age. These findings and accompanying screening data show that a large RCT of $O_2$ therapy is feasible. However, challenges associated with randomisation and data collection should be addressed in any future trial design.

## Introduction

Asthma is the most common chronic disease of childhood. In the UK, there is a person with asthma in one in five households and 1.1 million children are currently receiving treatment for this condition [1,2]. Oxygen ($O_2$) is a mainstay of treatment for acute severe or life-threatening asthma. There are physiological reasons why $O_2$ administered during an acute attack should be warmed and humidified [3]. BTS guidelines state that it is reasonable to use humidified $O_2$ for adult patients who need $O_2$ for longer than 24 hours or who report upper airway discomfort due to dryness (Evidence Grade D) [4], there is no such guidance for children. Currently, most children (and adults) with acute asthma receive cold (15˚C), dry (un-humidified) $O_2$ from bedside wall outlets as soon as they arrive in the Accident and Emergency (A&E) department. If hospitalised, they may or may not receive humidified $O_2$ depending on their local hospital asthma guidelines. There are few studies to support the use of humidified $O_2$ use in the acute setting for asthma or any respiratory condition and no randomised controlled trials. Over recent years, high flow nasal cannula $O_2$ has crept into the management of children with severe respiratory distress (not just asthma) in hospitals throughout the UK with a limited evidence base to support its use.

The HUMOX trial was performed to understand whether a future trial comparing different methods of administrating $O_2$ to children with severe asthma is feasible with regards to recruitment and retention, participant acceptability and adherence to the protocol.

## Methods

### Study design and participants

A 'Pilot study' design was used for this trial to determine whether a larger scale study could feasibly be carried out in the future. We did not intend to conduct hypothesis testing and make inferential conclusions regarding our results but rather evaluate the various processes involved in the trial, such as randomisation, recruitment and retention [5].

This multi-centre, open-label, parallel, pilot RCT recruited participants aged between 2–16 years with severe asthma according to the BTS and Scottish Intercollegiate Guidelines Network (SIGN) asthma guidelines [6] attending A&E Departments at four sites in the UK. To provide pilot data on ease of recruitment in both secondary and tertiary care, one large paediatric teaching hospital (Alder Hey Children's Hospital, Liverpool) and three district general hospitals (Royal Lancaster Infirmary, Warrington and Halton, and Countess of Chester) were selected. Exclusion criteria included requiring admission to intensive care, other respiratory disease or any other significant underlying medical problem.

The trial compared three ways of administering $O_2$ (heated humidified $O_2$, cold humidified $O_2$ or standard $O_2$). It was not possible to blind participants or any members of the trial team.

### Trial interventions

**Heated humidified $O_2$.** Heated humidified $O_2$ was delivered by a Fisher Paykel MR850 humidifier and a RT408 $O_2$ Therapy System through a System face-mask (No 1120 or 1100

depending on participant size). The humidifier was set to a temperature of 31˚C and the percentage inspired $O_2$ was titrated to maintain the participant's $O_2$ saturations above 92%. The humidifier was filled with sterile water with the levels monitored and topped up as necessary.

**Cold humidified $O_2$.** Cold humidified $O_2$ was given through an inter-surgical humidifier nebuliser, inserted into a bottle of sterile water and attached to wall-mounted low flow $O_2$. Elephant tubing was used to connect the nebuliser device to the participant's face-mask. Up to 60% $O_2$ was titrated to maintain the participant's $O_2$ saturations above 92%. If the participant required more than 60% $O_2$, a Rusch multi-fit nebuliser with BOC adapter was used in the same way.

**Standard $O_2$.** Standard cold (15˚C), dry (un-humidified) $O_2$ was given directly from the wall at the participant bedside via a non-rebreather mask. Once the participant required less than 10L $O_2$ (approximately 50% $FiO_2$), they were changed to nasal cannula.

## Randomisation

Stratified block randomisation (age (2–5 years and 6–16 years) and centre, random block sizes of 3 and 6) using a ratio of 1:1:1, was used within a computer generated list prepared by an independent statistician. Allocation concealment was ensured using sequentially numbered opaque, sealed envelopes. Regular checks were conducted on the envelopes to ensure that they were being used in the correct order and had not been tampered with.

Randomisation took place after completion of the screening phase and the initial nebulised treatment. The participant was re-assessed by the treating clinician and if they still required $O_2$ and fulfilled the entry criteria then the clinician/research nurse would take consent and randomise them by opening the next consecutive numbered envelope.

## Consent

The parent or legal representative of the child had an interview with the investigator, or a designated member of the investigating team, during which they were given the opportunity to understand the objectives, risks and inconveniences of the trial and the conditions under which it was to be conducted. They were provided with written information and contact details of the local study personnel should they require further information. Due to the nature of the study and the requirement to provide prompt treatment in an emergency setting, there was a short window of 90 minutes available for obtaining informed written consent in the A&E department/Paediatric Assessment Unit. Simplified written information was available for children 6–11 years, those aged 12–16 years and written assent was obtained when possible.

## Procedures

Participants commenced three 'back-to-back' nebulised salbutamol treatments with or without ipratropium bromide. Contemporaneously, parents/guardians were provided with study information documents and a screening assessment was undertaken. If they still required $O_2$ to maintain saturations ≥92%, then they could be randomised and if not they were treated as per standard guidelines.

Trial treatment began as soon as possible after the initial nebuliser treatment had concluded and initial assessments had been performed.

Following randomisation, trial participants were assessed at pre-specified time intervals (2, 4, 6, 8 and 12 hours and then every 6 hours following the start of the allocated intervention) for as long as they required $O_2$ (and until discharge). Data were collected on time taken for nebulised treatment to be definitively stepped down from randomisation to 1 hourly, 2 hourly

and 4 hourly treatments, and to salbutamol treatment delivered by metered dose inhaler and large volume spacer.

Adverse events were only reported where the causal relationship to the trial treatment had been assessed by the investigator to be related.

Prior to discharge the participant's parent or guardian (and participant if appropriate) was asked to consider what they thought were meaningful outcome measures for studies in acute asthma for the future and three months following discharge they were asked about their child's respiratory symptoms since discharge.

## Outcome measures

**Feasibility outcomes.**   Outcomes were not classified as primary or secondary. They were identified as relevant and important in a previous exercise involving consumers and paediatricians. However, during that exercise, the relative importance of these outcomes was not assessed. In this trial, the following outcomes were examined; length of time in $O_2$, time until treatment 'stepped down' to hourly, two-hourly and four-hourly nebulised therapy, difference in $O_2$ saturation in air after entry into the study, changes in Asthma Severity Score (ASS) [7], Paediatric Respiratory Assessment Measure (PRAM) [8], number of Salbutamol and Ipratropium Bromide nebules required by each participant following randomisation, requirement for escalation of treatment, adverse events, tolerability and length of stay in hospital.

Data using the Liverpool Respiratory Symptom Questionnaire (LRSQ) [9] were collected three months post discharge.

## Statistical analysis

**Sample size.**   A pragmatic sample size of 90 (30 in each of the three groups) was used [10].

**Data analysis.**   A statistical analysis plan was written prior to the analyses of the data [11]. All statistical analyses were conducted using SAS® V9.3 (SAS Institute, Cary, NC, USA).

Baseline data were described using summary statistics. Hypothesis testing were not carried out, rather data were summarised using summary statistics and 95% confidence intervals. Data were analysed using the intention to treat approach. As an aid to identifying potential outcome measures for a future trial, the proportion of missing data was assessed and there was no imputation.

## Approvals

The trial was approved by NRES committee North West Liverpool East on 01/11/2013 (13/NW/0738), given an International Standard Registered Clinical/social Number (62616194), sponsored by Alder Hey Children's NHS Foundation Trust and was overseen by an Independent Trial Steering Committee.

## Results

### Feasibility outcomes

The first participant was randomised on the 20[th] June 2014 and the final participant on the 23[rd] November 2016, the average recruitment was 2.2 participants per month. The trail recruitment finished at the end of the funding award. Between 5[th] June and 1[st] December 2016 a total of 675 participants were screened for inclusion into the study across four centres (Fig 1) and 66 were randomised across the three intervention arms (heated humidified $O_2$ n = 25, cold humidified $O_2$ n = 21 and standard $O_2$ n = 20).

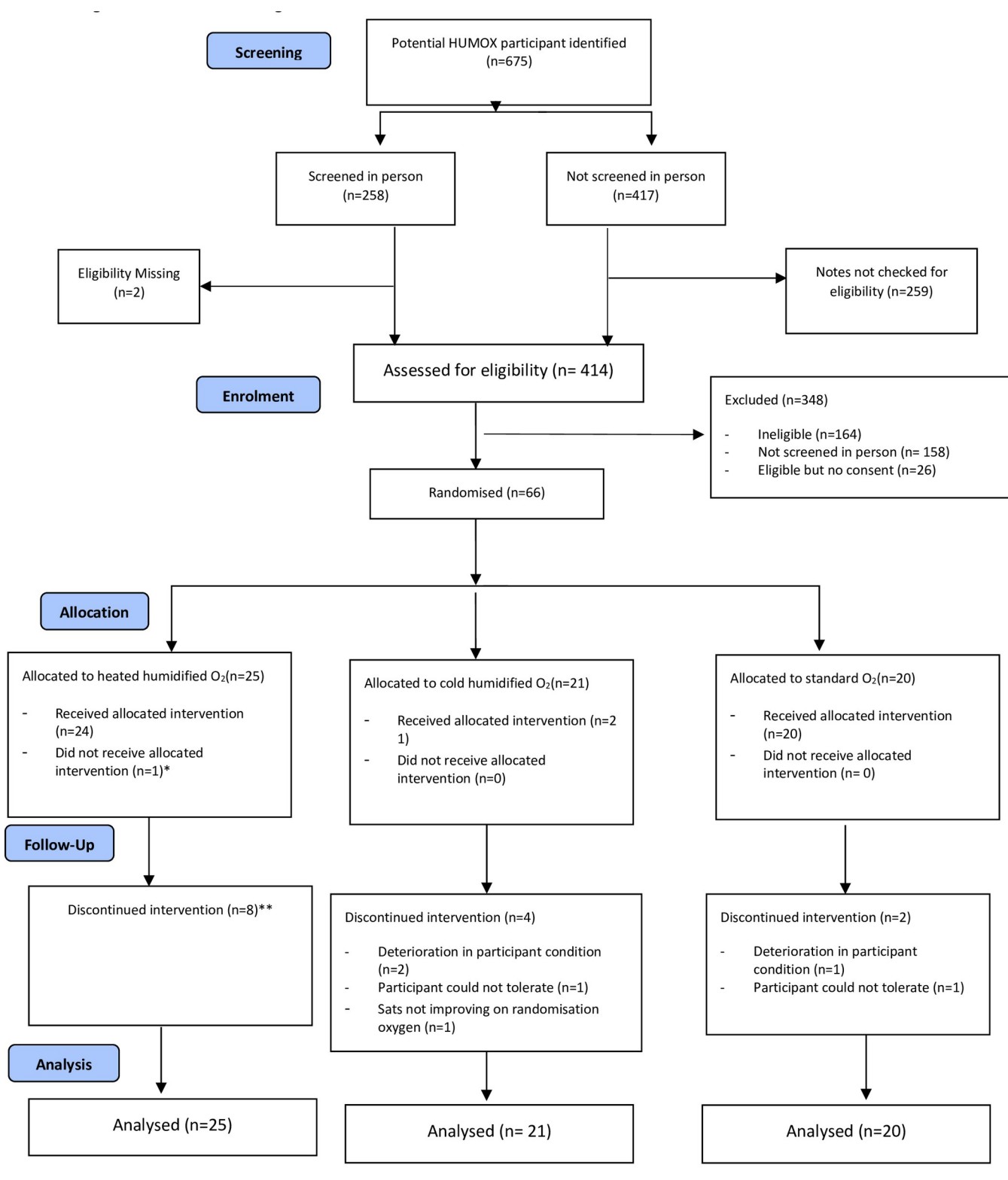

**Fig 1. CONSORT flow diagram.**

**Table 1. Baseline characteristics of individuals in the heated humidified $O_2$ (n = 25), cold humidified $O_2$ (n = 21) and standard therapy $O_2$ (n = 20) groups.**

| | | Heated humidified $O_2$ (n = 25) | Cold humidified $O_2$ (n = 21) | Standard therapy $O_2$ (n = 20) |
|---|---|---|---|---|
| **Age (years)** | Mean (SD) | 5.55 (2.6) | 4.81 (2.3) | 5.11 (2.0) |
| **Gender: n (%)** | Female | 9 (36%) | 11 (52%) | 5 (25%) |
| | Male | 16 (64%) | 10 (48%) | 15 (75%) |
| **ASS** | Mean (SD) | 5.88* (1.1) | 5.79** (0.8) | 6.11** (1.2) |
| **Age of asthma onset (years)** | Mean (SD) | 2.83 (1.9) | 2.00* (1.2) | 3.30 (2.2) |
| | Undiagnosed | 19 (76%) | 13 (65%) | 10 (50%) |
| **Previous admissions for asthma: n (%)\*** | 0 | 10 (40%) | 6 (30%) | 12 (60%) |
| | 1–4 | 13 (52%) | 11 (55%) | 2 (10%) |
| | >4 | 2 (8%) | 3 (15%) | 6 (30%) |
| **Time since previous admission (months)** | Median (IQR) | 3.59 (0.4, 48.0) | 4.00 (0.7, 30.0) | 19.00 (6.0, 58.4) |
| | Missing | 10 | 8 | 12 |
| **Allergy History: n (%)\*** | None | 16 (64%) | 10 (48%) | 7 (35%) |
| | Hay fever | 4 (16%) | 3 (14%) | 5 (25%) |
| | Eczema | 5 (20%) | 7 (33%) | 9 (45%) |
| | Food allergy | 1 (4%) | 2 (10%) | 4 (20%) |
| | Missing | 0 (0%) | 1 (100%) | 0 (0%) |
| **Length of current attack: n (%)\*** | Last 24 hrs | 14 (56%) | 8 (40%) | 8 (40%) |
| | Last 6 hrs or less | 3 (12%) | 1 (5%) | 2 (10%) |
| | Last few days | 8 (32%) | 11.00 (55%) | 10.00 (50%) |
| **Medication received prior to screening: n (%)\*** | No | 13 (52%) | 12 (60%) | 11 (55%) |
| | Not known | 1 (4%) | 1 (5%) | 0 (0%) |
| | Yes | 11 (44%) | 7 (35%) | 9 (45%) |

\* = 1 value missing

\*\* = 2 values missing.

There were 14 participants who discontinued their allocated treatment prematurely (see Fig 1). One participant in the heated humidified $O_2$ group did not start their allocated treatment and withdrew from the trial. Two participants (standard $O_2$ therapy) withdrew during their allocated treatment because their clinical condition deteriorated. A complete list of reasons for discontinuation is given in S1 Table.

## Baseline characteristics

Table 1 shows baseline characteristics prior to randomisation.

## Outcomes

**Length of stay in hospital (hours).** The median (IQR) was lower in the standard $O_2$ group (37.94 (29.1)) compared to that in both the heated humidified $O_2$ (52 (35.4)) and cold humidified $O_2$ (49.1 (29.7)) groups.

**Length of time on oxygen (hours).** The median (IQR) on $O_2$ was very similar in the heated humidified $O_2$ (13.6 (14.9)) and cold humidified $O_2$ (13.1 (14.9)) groups, whereas it was over two hours more in the standard $O_2$ group (15.9 (9.4)).

**ASS and PRAM.** The mean change from baseline in ASS during the first two hours was similar between the three groups (See Table 2). At six hours of treatment, the proportion of participants that had finished their treatment or had missing data rose to nearly 50% in all the groups making interpretation of data past this time point very difficult. The number of ASS

**Table 2. Change from baseline of ASS until 12 hours post-randomisation.**

| | Baseline | 2 hours Mean (SD) | Mean Difference (95% CI) | 4 hours | Mean Difference (95% CI) | 6 hours | Difference | 8 hours | Difference | 12 hours | Difference |
|---|---|---|---|---|---|---|---|---|---|---|---|
| **Heated humidified** $O_2$ | N = 24 5.88 (1.10) | N = 21 5.67 (1.4) | -0.3 (-0.8, 0.3) | N = 18 5.17 (0.79) | -0.8 (-1.4, -0.3) | N = 17 5.06 (1.14) | -0.9 (-1.7, -0.2) | N = 12 4.67 (0.89) | -1.5 (-2.3, -0.7) | N = 6 3.00 (1.53) | -2.0 (-3.5, -0.5) |
| **Cold humidified** $O_2$ | N = 19 5.79 (0.80) | N = 14 5.29 (1.2) | -0.4 (-1.0, 0.3) | N = 11 5.73 (1.42) | 0.0 (-1.0, 1.0) | N = 9 9.00 (1.27) | -0.2 (-1.8, 1.4) | N = 8 4.00 (1.20) | -1.8 (-3.0, -0.5) | N = 5 3.80 (1.30) | -2.0 (-3.8, -0.2) |
| **Standard** $O_2$ **therapy** | N = 18 6.11 (1.20) | N = 20 5.41 (1.28) | 0.6 (-1.2, 0.0) | N = 15 5.20 (1.66) | -0.6 (-1.2, -0.1) | N = 10 4.20 (1.48) | -1.7 (-2.7, -0.7) | N = 10 4.30 (1.89) | -1.8 (-3.4, -0.2) | N = 6 3.83 (2.56) | -2.7 (-5.2, -0.1) |

assessments that were missing or not assessed was greatest between the daily hours of 00:00 and 07:59 when there were fewer staff available to take measurements (see S1 Fig).

The first oversight committee meeting (held on the 23rd October 2015) noted that there was a significant amount of missing data for PRAM (S2 Fig), and so recommended that further collection should not continue. An amendment was then made to the protocol to remove PRAM as an outcome (Version 4.0 23rd November 2015).

**Time until treatment 'stepped down'.** Pooling data from all participants, the total mean (SD) time taken for nebulised treatment to be definitively stepped down from randomisation to 1 hourly, 2 hourly and 4 hourly was 2.1 (4.7) hours, 8.8 (8.2) hours and 14.5 (17.5) hours respectively. The mean (SD) time between randomisation and the start of inhaled salbutamol treatment delivered by metered dose inhaler and large volume spacer device for heated humidified $O_2$, cold humidified $O_2$ and standard $O_2$ therapy groups was 35.1 (28.2) hours, 32.7 (20.1) hours and 25.6 (14.3) hours respectively.

**Difference in oxygen saturation.** The mean (SD) change in baseline $O_2$ saturations in air was similar between all three groups. However, change in baseline saturations tended to be lower in the standard $O_2$ group for most time points over this period (see S3 Fig).

**Salbutamol and Ipratropium bromide usage.** The median (IQR) number of salbutamol nebules required in each of the three treatment groups was similar (12.0 (11.0) in the heated humidified $O_2$, 10 (5) in the cold humidified $O_2$ and 9.50 (7.50) in the standard $O_2$ therapy).

The median (IQR) number of ipratropium bromide nebules required were 3.0 (4.5), 2.0 (3.0) and 2.5 (4.0) for the heated humidified $O_2$, cold humidified $O_2$ and standard $O_2$ therapy groups respectively.

**Escalation of treatment.** The number of participants who required escalation of treatment was greater in the heated humidified $O_2$ group (16 (66.7%)) (cold humidified $O_2$ (7 (33%)) and standard $O_2$ group (11 (55%))).

**Liverpool Respiratory Symptom Questionnaire and parental assessment of asthma outcomes.** The symptom scores for each of the domains of the LRSQ were similar across the three treatment groups with the standard $O_2$ therapy group having a slightly higher overall score than both the cold and heated humidified $O_2$ groups (see Fig 2). The mean score given by parents for each of the suggested asthma outcomes was similar overall and between each of the treatment groups (see S4 Fig).

**Adverse events.** There were no serious adverse events. There were nine adverse reactions reported (all mild in severity), eight of which were related to participants being unable to tolerate their $O_2$ (heated humidified $O_2$ n = 6 and cold humidified $O_2$ n = 1 and standard $O_2$ group

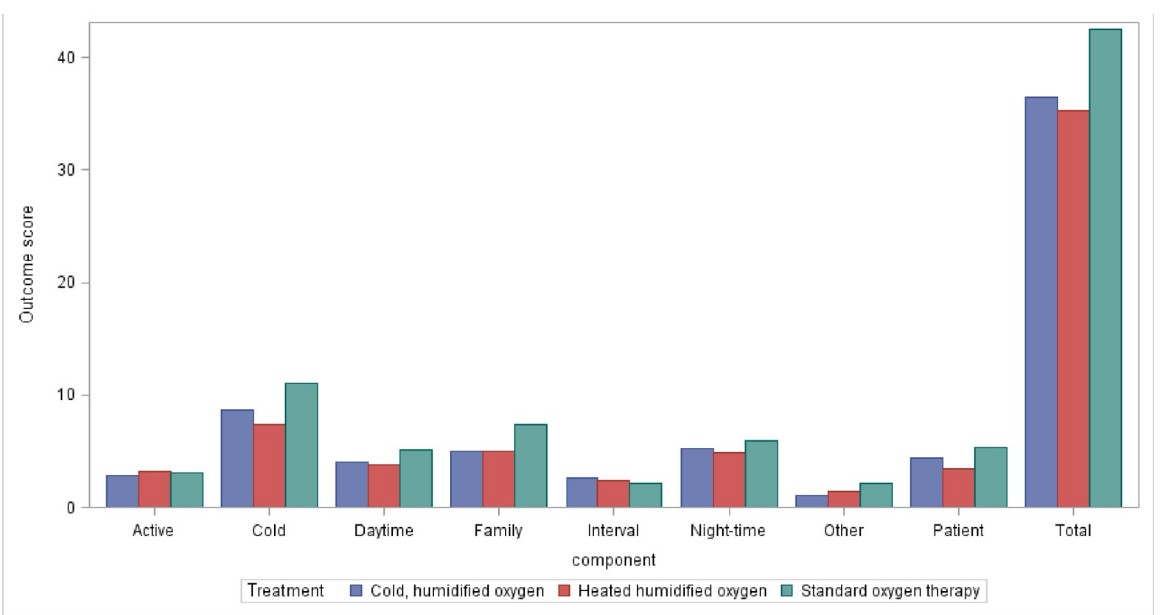

**Fig 2. Respiratory symptom score by component and treatment group.**

n = 1), and the remaining adverse reaction was related to the patient being unable to maintain $O_2$ saturations above 92% (heated humidified $O_2$).

## Discussion

This is the first RCT to compare humidified and standard $O_2$ therapy in acute severe asthma. Data on recruitment, consent and randomisation, treatment and outcome measure suitability have been generated to inform future large definitive clinical trials on interventions for acute severe asthma in children and young people.

This trial highlights several practical issues pertinent to future trial design in this patient group. Firstly, although more than half the recruited cohort had had at least one previous respiratory admission, most (65%) didn't have a formal diagnosis of asthma. As such, it would have been very difficult to prospectively recruit to this study by targeting otherwise well children attending outpatient clinics with asthma/viral induced wheeze. Secondly, predictions for recruitment to future similar studies cannot be based solely on numbers of children attending A&E requiring $O_2$. For those children attending A&E who were hypoxic, only 60% still needed $O_2$ after initial triple nebuliser treatment. Thirdly, recruitment was most successful in centres where there was a dedicated research nurse assigned to the study. Flexible recruitment both at weekends and during the night was also desirable given that nearly half those attending hospital with severe asthma arrived between 5pm and midnight. Lastly, even though information sheets were given at a stressful time, consent rates were high (72%).

Despite offering incentives (£50 gift vouchers in prize draws), it was challenging to get nursing and medical staff to undertake an online training course to calculate the PRAM score. Ultimately, PRAM was removed from the clinical report following poor data completion. This was disappointing as PRAM has been extensively validated as an outcome measure in children and young people between 2–17 years with acute severe asthma in North America [8,12] but there have been relatively few studies using it in Europe and none in the UK. As for ASS, completion rates were better but still only half the data was complete between 00:00 and 07:59, likely reflecting the busy workloads of clinical staff out-of-hours.

In the time it has taken to complete the trial, several issues relevant to future trial design have arisen. Firstly, the EU clinical trials regulation published in 2014 and approved in the UK in 2016, would now classify $O_2$ as a drug. Secondly, clinical practice regarding the acute management of children with severe respiratory distress of whatever cause has changed, particularly in the UK. High flow heated humidified $O_2$ given via nasal cannula therapy ($HFNCO_2$) has 'crept' into the clinical management of children, often with very little evidence to support its use. Largely because of this we would see the next steps being a comparative trial of $HFNCO_2$ and standard $O_2$ therapy, a feasibility study for which is already in development [13].

The number of participants required for a future trial were calculated for length of time on $O_2$ and length of time in hospital. Data for both these outcomes were not normally distributed and therefore the methods described by O'Keeffe [14] were used. A sample size calculation to detect a minimum clinically important difference of 20% for the length of time on $O_2$ was undertaken and would require a sample size of 214 in each group and to detect the same difference for the length of time in hospital would require 114 in each group.

Further work should be undertaken on what outcomes are important to patients, parents and healthcare providers for acute severe asthma and on minimally clinical important differences. How important is a 20% reduction in time requiring $O_2$ or time in hospital equating to ~3 and 8 hours respectively to stakeholders and particularly funders, when lengths of stay are generally so short? Treatment escalation (in the form of need for HDU/PICU, IV aminophylline/salbutamol) may have better potential as a primary outcome given that it was not uncommon (particularly in those receiving heated humidified $O_2$) and may be more important to key stakeholders.

Given that $HFNCO_2$ has crept into the management of children with severe respiratory distress in hospitals throughout the UK, any such future trial would likely incorporate this intervention rather than heated or cold humidified $O_2$ by face-mask.

## Supporting information

**S1 Checklist.**
(DOC)

**S1 Fig. Number of ASS assessments missing/not assessed/assessed at different time points.**
(DOCX)

**S2 Fig. Number of PRAM assessments missing/not assessed/assessed at different time points.**
(DOCX)

**S3 Fig. Change in oxygen saturation by treatment group in the first 24 hours.**
(DOCX)

**S4 Fig. Parental assessment of outcomes.**
(DOCX)

**S1 Table. Reasons for discontinuation of allocated intervention.**
(DOCX)

**S1 File.**
(PDF)

**S2 File.**
(PDF)

**S3 File.**
(PDF)

**S4 File.**
(PDF)

## Acknowledgments

The authors would like to thank all the participants and families who gave their time and took part in HUMOX and all the research staff at the sites that took part in the study. The authors would also like to thank the independent members of the Trial Steering Committee (Chair: Dr Clare Murray and Dr Chris Sutton), Mrs Tracy Moitt for oversight at the Clinical Trials research centre and Dr Steven Lane for statistical advice during the early phase of the trial.

Heated humidifiers for the study were provided from the manufacturing company Fisher Paykel and then at the end of the study were kindly donated to Alder Hey Children's NHS Foundation Trust.

## Author Contributions

**Conceptualization:** Paul S. McNamara, Vanessa Compton, Matthew Peak.

**Formal analysis:** Dannii Clayton, Ashley P. Jones.

**Funding acquisition:** Paul S. McNamara, Vanessa Compton, Matthew Peak.

**Investigation:** Vanessa Compton, Matthew Peak, Janet Clark.

**Methodology:** Paul S. McNamara, Dannii Clayton, Matthew Peak, Janet Clark, Ashley P. Jones.

**Project administration:** Paul S. McNamara, Caroline Burchett, Janet Clark.

**Supervision:** Paul S. McNamara.

**Writing – original draft:** Paul S. McNamara, Ashley P. Jones.

**Writing – review & editing:** Paul S. McNamara, Dannii Clayton, Caroline Burchett, Vanessa Compton, Matthew Peak.

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
