## [Decision Letter · Decision Letter 0]

19 Nov 2020

PONE-D-20-30805

Humidified and standard oxygen therapy in acute severe asthma in children (HUMOX): a pilot randomised controlled trial

PLOS ONE

Dear Dr. Jones,

Thank you for submitting your manuscript to PLOS ONE. After careful consideration, we feel that it has merit but does not fully meet PLOS ONE’s publication criteria as it currently stands. Therefore, we invite you to submit a revised version of the manuscript that addresses the points raised during the review process.

It is somewhat astonishing that you would use the worst possible randomization method even in this day and age, when so much has been published about the shortcomings of permuted block randomization, even with varying block sizes.  What possible justification can there be for this?  Before you respond, please carefully review the following:

Berger, VW, Ivanova, A, Deloria-Knoll, M (2003). “Minimizing Predictability while Retaining Balance through the Use of Less Restrictive Randomization Procedures”, *Statistics in Medicine*
**22**, 19, 3017-3028.

Berger, VW (2005). “Selection Bias and Covariate Imbalances in Randomized Clinical Trials”, John Wiley & Sons, Chichester.

Berger, VW (2006). “Do Not Use Blocked Randomization”, *Headache*
**46**, 2, 343.

Berger, VW (2006). “Varying Block Sizes Does Not Conceal the Allocation”, *Journal of Critical Care*
**21**, 2, 229.

Berger, VW (2006). “Misguided Precedent Is not a Reason To Use Permuted Blocks”, *Headache*
**46**, 7, 1210-1212.

Berger, VW (2015). “Failure To Look Beyond Blocks Is a Mistake”, *Methods of Information in Medicine*
**54**, 3, 290.

Berger, VW (2015). “Concealing the Block Sizes Is Not Sufficient”, *Clinics in Orthopedic Surgery*
**7**, 422-423.

Berger, VW Agnor, RC, Bejleri, K (2016). “Comparing MTI Randomization Procedures to Blocked Randomization”, *Statistics in Medicine*
**35**, 5, 685-694.

Zhao, WL and Berger, VW (2017). “Better Alternatives to Permuted Block Randomization for Clinical Trials with Unequal Allocation”, *Hematology*
**22**, 1, 61-63.

Zhao, WL, Berger, VW, Yu, Z (2017). "The Asymptotic Maximal Procedure for Subject Randomization in Clinical Trials", *Statistical Methods in Medical Research*
**27**, 7, 2142-2153.

Especially in an unmasked trial, this is inexcusable.  It is stated that:

"Allocation concealment was ensured using sequentially numbered opaque, sealed envelopes".

How would sealed envelopes in any way, shape, or form ensure allocation concealment?  Short answer:  They don't.  The reality is that allocation concealment is rendered impossible by the combination of the lack of masking and the fatally flawed randomization method used.  See:

Berger, VW (2005). “Is Allocation Concealment a Binary Phenomenon?”, *Medical Journal of Australia*
**183**, 3, 165.

Berger, VW Do, AC (2010). “Allocation Concealment Continues To Be Misunderstood”, *Journal of Clinical Epidemiology ***63**, 4, 468-470.

What is the plan for dealing with missing data?

How will the data be analyzed?

Why are there no p-values in Table 1?  See:

Berger, VW (2009). “Do Not Test for Baseline Imbalances Unless They Are Known To Be Present?”, *Quality of Life Research*
**18**, 399.

Berger, VW (2010). “Testing for Baseline Balance: Can We Finally Get It Right?”, *Journal of Clinical Epidemiology ***63**, 8, 939-940.

We look forward to receiving your revised manuscript.

Kind regards,

Vance Berger

Academic Editor

PLOS ONE

Journal Requirements:

2. Please ensure that all figures and tables have been corrected referenced in-text.

3. In your Methods section, please provide additional information about the participant recruitment method and the demographic details of your participants. Please ensure you have provided sufficient details to replicate the analyses such as: a) a description of how participants were recruited, and b) descriptions of where participants were recruited and where the research took place.

4. Please provide additional details regarding participant consent. In the ethics statement in the Methods and online submission information, please ensure that you have specified (1) whether consent was informed and (2) what type you obtained (for instance, written or verbal, and if verbal, how it was documented and witnessed). Since your study included minors, please also state whether you obtained consent from parents or guardians.

7. Your ethics statement should only appear in the Methods section of your manuscript. If your ethics statement is written in any section besides the Methods, please delete it from any other section.

8. Please ensure that you refer to Figures 1 and 2 in your text as, if accepted, production will need this reference to link the reader to the figure.

9. We note you have included a table to which you do not refer in the text of your manuscript. Please ensure that you refer to Table 2 in your text; if accepted, production will need this reference to link the reader to the Table.

10. Please include captions for your Supporting Information files at the end of your manuscript, and update any in-text citations to match accordingly. Please see our Supporting Information guidelines for more information: http://journals.plos.org/plosone/s/supporting-information.

Reviewers' comments:

Reviewer's Responses to Questions

**Comments to the Author**

1. Is the manuscript technically sound, and do the data support the conclusions?

Reviewer #1: Partly

2. Has the statistical analysis been performed appropriately and rigorously? 

Reviewer #1: No

3. Have the authors made all data underlying the findings in their manuscript fully available?

Reviewer #1: Yes

4. Is the manuscript presented in an intelligible fashion and written in standard English?

Reviewer #1: Yes

5. Review Comments to the Author

Reviewer #1: The manuscript entitled ‘Humidified and standard oxygen therapy in acute severe asthma in children (HUMOX): a pilot randomised controlled trial’ with the aim to examine the feasibility of humidified O2 (heated humidified or cold humidified ) to standard O2 in children with severe acute asthma and to obtain data on recruitment, tolerability and outcome measure stability.

The manuscript can be further improved based on the comments below and requires thorough proofreading.

Abstract

Page 3 Line 39-40, the sentence ‘ (shortest for standard O2, 37.9(29.1) hours) and 14.6 (14.2) hours (shortest for cold humidified O2, 13.1(14.9) hours) respectively’ to be revised and to state for all groups with their respective Median ±IQR.

Page 3 Line 41, for ‘inhaled treatment was 31.4 (22.2) hours, the group name to be stated. Likewise to include other groups readings.

Introduction

Page 5, Introduction was too short. More information to be provided.

Methods

Recruitment

Page 6, more information to be provided on how the subjects were recruited (as illustrated in Figure 1) in the method section.

Randomization

Page 7 Line 95-96, more description to be provided for this sentence ‘Stratified block randomisation (age (2-5 years and 6-16 years) and centre, block sizes of 3 and 6)’

Outcome measures

Page 8 Line 129, full name for abbreviation ASS (Asthma Severity Score) to be stated.

Data analysis

Page 9 Line 141, statistical software including version and publisher name which was used to perform the descriptive statistics to be stated.

More information on missing data to be provided in terms of percentage, pattern etc.

Results

Page 10 Line 157, incomplete sentence.

Page 10 Line 158-162 and 164-168, content were similar and repeated.

Page 11-12, Line 180-181, sentence missing.

Page 13 Line 194-199, separate groups findings to be provided.

Page 10 & 14 Line 158, Line 164, Line 216, there were many ‘Error! Reference source not found:’

There were many missing data/not assessed and this needs to be discussed.

Table 1, the title can be expanded. At least 1 decimal point for the percentage figures. In the table footnote, it was stated ‘***=3 values missing’ but was not found in the table. Range to be replaced with IQR.

Table 2, the readings at baseline, 2-hour, 4-hour, 6-hour, 8-hour and 12-hour to be provided before deriving the mean difference. Intent to treat and n to be stated.

Figure 1, the assessment period to be incorporated in and intention to treat to be stated.

Figure 2, n to be stated. There were two titles. Title in the graph to be removed.

6. PLOS authors have the option to publish the peer review history of their article (what does this mean?). If published, this will include your full peer review and any attached files.

Reviewer #1: No

---

## [Author Response · Author response to Decision Letter 0]

8 Jun 2021

Response to all comments have been uploaded.

---

## [Decision Letter · Decision Letter 1]

6 Dec 2021

PONE-D-20-30805R1Humidified and standard oxygen therapy in acute severe asthma in children (HUMOX): a pilot randomised controlled trialPLOS ONE

Dear Dr. Jones,

Thank you for submitting your manuscript to PLOS ONE. After careful consideration, we feel that it has merit but does not fully meet PLOS ONE’s publication criteria as it currently stands. Therefore, we invite you to submit a revised version of the manuscript that addresses the points raised during the review process.

We look forward to receiving your revised manuscript.

Kind regards,

Andres Azuero, Ph.D., MBA

Academic Editor

PLOS ONE

Journal Requirements:

Additional Editor Comments:

Dr. Jones - I have read the revision of your manuscript as well as feedback from two pulmonologists who reviewed it. I have also read the reviews from your initial submission. In my opinion, the presentation of the research is in line with what can be expected of a pilot project: the focus is on feasibility and acceptability, not inference. The goal is to try out the procedures and logistics and see what works and what doesn’t so that problems in a large confirmatory trial are avoided. The outcome data themselves are secondary, and whatever conclusion of benefit or not, apply to the sample only; no statement should be made about generalizability of results and no formal testing should be conducted, although uncertainty can be expressed in the form of confidence intervals or credible intervals.

Regardless, looking at the reviews from the original submission and the revision, a common theme is that reviewers had a difficult time seeing this study as a pilot and tried to interpret it, at least partially, as if it was a confirmatory study, which is clearly not, and therefore the concerns about baseline imbalances, inferential testing, and statistical power. In addition to that, the prior academic editor, who has studied in-depth the nuances of randomization procedures, expressed concern about the randomization procedure and how it was implemented.

Therefore, to avoid confusion and over-interpretation of results, the following is needed:

1) A statement in the abstract along the lines of:

“Because of the small sample size of this pilot, conclusions cannot be extrapolated beyond the study sample.” So that from the beginning readers are not looking for inferential conclusions.

2) A paragraph in the methods section explaining to readers the purpose of a pilot study such as yours. In addition to your reference 9 (Lancaster et al, 2004), consider using the following:

https://www.nccih.nih.gov/grants/pilot-studies-common-uses-and-misuses

https://www.ncbi.nlm.nih.gov/pmc/articles/PMC3081994/

https://www.ncbi.nlm.nih.gov/pmc/articles/PMC4917389/

3) A paragraph in the discussion about what you will do to improve the randomization in a future confirmatory trial. For instance, increase the number of random block sizes to 3, 6, 9, 12, and 15, or use a different randomization method altogether, and implement a computerized system (as opposed to the envelopes).

Minor fix: typo in abstract results: “[…]time (hours) taken to step down nebulised to inhaled treatment was 5.6 (14.3),” should be 35.6 (14.3).

Finally, please respond to the reviewer comments.

Reviewers' comments:

Reviewer's Responses to Questions

**Comments to the Author**

1. If the authors have adequately addressed your comments raised in a previous round of review and you feel that this manuscript is now acceptable for publication, you may indicate that here to bypass the “Comments to the Author” section, enter your conflict of interest statement in the “Confidential to Editor” section, and submit your "Accept" recommendation.

Reviewer #2: (No Response)

Reviewer #3: (No Response)

2. Is the manuscript technically sound, and do the data support the conclusions?

Reviewer #2: Yes

Reviewer #3: Yes

3. Has the statistical analysis been performed appropriately and rigorously? 

Reviewer #2: Yes

Reviewer #3: Yes

4. Have the authors made all data underlying the findings in their manuscript fully available?

Reviewer #2: Yes

Reviewer #3: Yes

5. Is the manuscript presented in an intelligible fashion and written in standard English?

Reviewer #2: Yes

Reviewer #3: Yes

6. Review Comments to the Author

Reviewer #2: The manuscript, "Humidified and standard oxygen therapy in acute severe asthma in children (HUMOX): a pilot randomized controlled trial" by Jones and colleagues examines the feasibility of comparing types supplemental oxygen in asthmatic children experiencing acute exacerbations. The authors identify several potential barriers to a full-scale investigation, including difficulties in recruiting volunteers to complete some of the desired outcome measures, and difficulties in achieving adequate overnight data collection. As such, this manuscript provides insights that will be useful to clinical investigators studying other clinically relevant questions. Therefore, this manuscript will be of interest to a fairly general audience. The manuscript has been significantly improved by the revisions. However, there are still some areas that are not clear in the current manuscript. These minor issues include the following:

1. The Procedures section specifies the duration of assessment to be "as long as they required O2 (and until discharge)". Do the authors mean that all the patients were followed until their discharge? Alternatively, this sentence could mean the patients were followed until they were either off O2 or had been discharged. This sentence should be rewritten to be more clear.

2. The Procedures mention that the parents or guardians provided recommendations regarding potential meaningful patient outcomes and post-A&E Department therapy. The details regarding these findings are provided in the Supplemental Material. However, no reference to this is made within the Procedures section. The authors should state that these findings are provided in the Supplemental Material appendix to make it easier for the interested reader to locate them.

3. The authors state that the outcomes were not designated as primary or secondary, but do not provide a rationale for doing that. The authors should provide a rationale as to why they chose to not designate outcomes as primary or secondary.

4. The authors should provide more details regarding how they arrived at 30 per group as their sample size. Did they conduct a power analysis? Was this based on the average number of children seen in the A&E Department for acute asthma exacerbations?

5. The Discussion outlines some of the barriers experienced by the authors in obtaining their data. While they report some of the approaches that were not successful, it would be helpful if they would include potential ways that they might be successful in collecting a more complete data set in a subsequent study. This is particularly important since the difficulties with missing data precluded the use of a clinically relevant assessment, the PRAM score.

Reviewer #3: In the table !, when you compare the three groups with parametric method do you use to write they are similar?

Can you explain why did not carried out hypothesis testing?

7. PLOS authors have the option to publish the peer review history of their article (what does this mean?). If published, this will include your full peer review and any attached files.

Reviewer #2: No

Reviewer #3: No

---

## [Author Response · Author response to Decision Letter 1]

22 Dec 2021

We thank the editor and reviewers for their helpful and constructive comments which have improved the manuscript. We have included a table with our responses point by point with this revision.

---

## [Editor Report · Decision Letter 2]

12 Jan 2022

Humidified and standard oxygen therapy in acute severe asthma in children (HUMOX): a pilot randomised controlled trial

PONE-D-20-30805R2

Dear Dr. Jones,

We’re pleased to inform you that your manuscript has been judged scientifically suitable for publication and will be formally accepted for publication once it meets all outstanding technical requirements.

Kind regards,

Andres Azuero, Ph.D., MBA

Academic Editor

PLOS ONE

---

## [Editor Report · Acceptance letter]

25 Jan 2022

PONE-D-20-30805R2 

Humidified and standard oxygen therapy in acute severe asthma in children (HUMOX): a pilot randomised controlled trial 

Dear Dr. Jones:

I'm pleased to inform you that your manuscript has been deemed suitable for publication in PLOS ONE. Congratulations! Your manuscript is now with our production department. 

Kind regards, 

on behalf of

Dr. Andres Azuero 

Academic Editor

PLOS ONE